# Targeting HDAC2-Mediated Immune Regulation to Overcome Therapeutic Resistance in Mutant Colorectal Cancer

**DOI:** 10.3390/cancers15071960

**Published:** 2023-03-24

**Authors:** Mariarosaria Conte, Annabella Di Mauro, Lucia Capasso, Liliana Montella, Mariacarla De Simone, Angela Nebbioso, Lucia Altucci

**Affiliations:** 1Department of Precision Medicine, University of Campania “Luigi Vanvitelli”, 80138 Naples, Italy; annabella.dimauro@unicampania.it (A.D.M.); lucia.capasso@unicampania.it (L.C.); angela.nebbioso@unicampania.it (A.N.); lucia.altucci@unicampania.it (L.A.); 2Pathology Unit, Istituto Nazionale Tumori-IRCCS-Fondazione G. Pascale, 80131 Naples, Italy; 3ASL NA2 NORD, Oncology Operative Unit, “Santa Maria delle Grazie” Hospital, 80078 Pozzuoli, Italy; liliana.montella@aslnapoli2nord.it; 4Stem Cell Transplantation Unit, Division of Hematology, Cardarelli Hospital, 80131 Naples, Italy; mariacarla.desimone@aocardarelli.it; 5BIOGEM, Institute of Molecular Biology and Genetics, 83031 Ariano Irpino, Italy; 6IEOS, Institute for Endocrinology and Experimental Oncology, CNRs, 80131 Napoli, Italy

**Keywords:** epigenetics, colorectal cancer, immune infiltrates, patient, mutation, genetic/epigenetic/immunological signature

## Abstract

**Simple Summary:**

Epigenetic processes contribute to the regulation of the immune system by activating multiple transcriptional changes, which in turn, are the product of immune cell reprogramming. Given that the deregulation of these mechanisms promotes cancer progression by altering the balance of genes controlling cell proliferation and death, the objective of this study was to identify a genetic/epigenetic/immunological colorectal cancer signature through a preliminary in silico analysis aimed at identifying the pathogenic causes of colorectal cancer associated with expression levels of histone deacetylase 2 (*HDAC2*) and two immune system regulators, class II major histocompatibility complex transactivator (*CIITA)* and beta-2 microglobulin (*B2M*), in a cohort of patients harboring a common dysregulation of these genes. We next extended the study by investigating a tissue microarray cohort of colorectal cancer patients from a diagnostic/prognostic perspective.

**Abstract:**

A large body of clinical and experimental evidence indicates that colorectal cancer is one of the most common multifactorial diseases. Although some useful prognostic biomarkers for clinical therapy have already been identified, it is still difficult to characterize a therapeutic signature that is able to define the most appropriate treatment. Gene expression levels of the epigenetic regulator histone deacetylase 2 (*HDAC2*) are deregulated in colorectal cancer, and this deregulation is tightly associated with immune dysfunction. By interrogating bioinformatic databases, we identified patients who presented simultaneous alterations in *HDAC2*, class II major histocompatibility complex transactivator (*CIITA)*, and beta-2 microglobulin (*B2M*) genes based on mutation levels, structural variants, and RNA expression levels. We found that *B2M* plays an important role in these alterations and that mutations in this gene are potentially oncogenic. The dysregulated mRNA expression levels of *HDAC2* were reported in about 5% of the profiled patients, while other specific alterations were described for *CIITA*. By analyzing immune infiltrates, we then identified correlations among these three genes in colorectal cancer patients and differential infiltration levels of genetic variants, suggesting that *HDAC2* may have an indirect immune-related role in specific subgroups of immune infiltrates. Using this approach to carry out extensive immunological signature studies could provide further clinical information that is relevant to more resistant forms of colorectal cancer.

## 1. Introduction

Most colorectal cancers (CRCs) are classified as sporadic forms and account for ~80% of all cases, while the remaining ~20% occur in the context of a hereditary syndrome [1]. Currently, colectomy is used to prevent or treat some morbid conditions that arise in the colon. For early-stage tumors, localized surgical resection is performed in about 50% of patients [2]. Targeted therapies aim to prolong survival, delay progression, reduce the volume of metastases, and improve overall quality of life [3]. CRC is often diagnosed at an advanced stage, when the disease has already spread to the lymph nodes and/or more distant organs. Since survival in CRC patients improves with early diagnosis, there is an urgent need for screening programs that are able to identify specific and early disease markers [4,5,6].

Microsatellite instability (MSI) is a characteristic phenotype of CRC [7]. MSI is usually associated with an unstable tumor phenotype, and this association was highlighted in a study comparing normal and tumor DNA sequences in CRC patients [8]. In 20% of cases, MSI in CRC is caused by the inactivation of one allele of a mismatch repair gene by a germline mutation and subsequent somatic inactivation of the second allele. The remaining 80% of CRC cases involve somatic inactivation of both alleles of the *MLH1* gene, which is involved in DNA mismatch repair via hypermethylation changes (Lynch syndrome), or they are hereditary nonpolyposis colon cancers [9]. MSI tumors lose beta-2 microglobulin *(B2M)* expression, which leads to a complete loss of HLA-I antigen presentation resulting in decreased immune surveillance [10]. As chemotherapy achieves better survival in CRC harboring MSI than in CRC with microsatellite stability (MSS) or low MSI, it is critical to identify the markers that are able to better stratify patients for treatment regimens, including adjuvant therapy [11,12]. CRCs are classified into four different consensus molecular subtypes (CMS1, CMS2, CMS3, and CMS4) based on gene expression assays and they are characterized by different immune signatures. CMS1 exhibits an overexpression of genes involving cytotoxic lymphocytes, CMS2 and CMS3 present a low immune signature, while CMS4 has an immunosuppressive signature. Based on these findings and given that CRC is a very heterogeneous tumor type with a high mutation rate, immunotherapeutic approaches may be more effective for specific cancer subtypes [13]. The risk factors associated with CRC can be both environmental and behavioral, while the complexity of the disease is the result of genetic mutations and epigenetic modifications. The timeline of different mutations can vary from patient to patient. Clinically, the identification of a particular mutation may have prognostic value, while epigenetic alterations impact several biological systems, including the immune system [14]. Histone deacetylases (HDACs) are chromatin regulators with epigenetic functions that control the acetylation/deacetylation status. Gene expression levels of *HDAC2*, a member of the class I HDAC family, are known to be deregulated in CRC [15]. Homozygous inactivation of *HDAC2* occurs in sporadic CRC with MSI and hereditary nonpolyposis colon cancer syndrome, suggesting that these tumors may develop in the absence of *HDAC2*, a hypothesis that could have major implications for cancer therapy [16]. *HDAC2* overexpression in CRC is responsible for adenoma–carcinoma progression [17], while *HDAC2* mutations are associated with treatment resistance [18]. Another fundamental aspect that characterizes CRC tumors with MSI is the distinction between driver and bystander mutations. Since *HDAC2* was observed to be subject to constant and adaptive mutations in CRC lines in vitro, it could be considered a bystander gene in CRC tumorigenesis. This may also explain why some CRCs have different susceptibilities to drug treatments. Drugs with epi-activity, such as HDAC inhibitors (HDACi), represent a promising therapeutic approach for the combination treatment of CRC [19,20,21]. In our previous studies [22,23], we identified the class II major histocompatibility complex transactivator (*CIITA)* gene as an important interactor of *HDAC2* and consequently, of major histocompatibility complex (*MHC*) class II regulatory genes, which contribute to CRC progression. Intriguingly, *B2M*, which is part of the MHC class I system, promotes numerous forms of resistance and is associated with tumor escape [24]. Several CRC tumors show similar elevated and/or deregulated *B2M* levels, which are linked to *HDAC2* and *CIITA* expression, although *HDAC2* mutations have not yet been well characterized. *B2M* and *CIITA* are tightly regulated at the transcriptional level by *HDAC2*. Since acetylation is a reversible and dynamic process, HDAC2 could use CIITA and B2M as substrates in a deacetylation reaction and thus antagonize their activity. Depending on its expression level, *HDAC2* can positively or negatively control the transcriptional activity of *CIITA* and *B2M*, which, in turn, can be differently regulated within the tumor microenvironment. Based on these observations, we aimed to identify a genetic/epigenetic/immunological signature of CRC and provide additional insight into therapeutic resistance that could lead to improved patient stratification, with a potential impact on clinical practice. We set up a targeted study focusing on *HDAC2* aberrations and *HDAC2*-regulated genes associated with immune dysfunction in CRC. We used an in silico analysis to explore the distribution of immune infiltration in *HDAC2*, *B2M*, and *CIITA* genes and their correlation with mutational status. We also investigated the percentage of common mutations, structural variants, mRNA expression, and amplifications in the three genes of interest in a subset of patients selected from a total of 594 CRC patients/samples obtained from the Pan Cancer Atlas of The Cancer Genome Atlas (TCGA). In some of these patients, mutations/deregulations of common expression levels involved all three genes simultaneously, while in others only two were involved. We analyzed the data from patients harboring these alterations and compared the findings with those obtained from tissue microarrays (TMAs) in a cohort of CRC patients. We found elevated levels of *HDAC2* in TMA patients associated with *B2M* and *CIITA* expression. The aim of this study was to provide additional information about CRC physiopathology and investigate the possibility of adopting more precise diagnostic criteria in order to enhance the use of immune-based therapeutic interventions in the clinic.

## 2. Materials and Methods

### 2.1. Systematic Analysis of HDAC2-, CIITA-, and BM2-Dependent Immune Infiltrates across CRC Samples

The TIMER database (https://cistrome.shinyapps.io/timer/ accessed on 29 December 2022) was interrogated to investigate tumor-infiltrating immune cells in colon adenocarcinoma (COAD) tumors with different somatic copy number alterations (SCNA) of *CIITA*, *HDAC2*, and *B2M* genes. We compared immune infiltration distribution by looking at the SCNA status of the genes of interest across TCGA cancer types. We estimated SCNA information from copy number segmentation profiles at gene level, including deep deletions, arm-level deletions, diploid/normal status, arm-level gains, and high amplifications. We then analyzed the deep deletion or high amplification alteration status of the genes of interest for comparison with the diploid/normal status. We also used the outcome module to explore the clinical relevance of immune infiltrate subsets based on cumulative survival. In addition, we evaluated the correlation between CD8^+^/CD4^+^ T cells and interrogated genes. We then analyzed differential infiltration levels of CD4^+^ Th1 and Th2 T cells in mutated forms of the selected genes in COAD using the xCell platform. We also used xCell to explore differential infiltration levels of central and effector memory CD8^+^ T cells in these genes. The “purity adjusted” option was selected to highlight the relationship between gene expression and tumor purity (the percentage of cancer cells in a sample) as well as the relationship between gene expression and immune cell type.

### 2.2. HDAC2 and CIITA Correlation by DepMap/CCLE

We investigated the correlation between HDAC2 and CIITA in 73 different colon cancer cell lines by interrogating the DepMap/CCLE database (https://depmap.org/portal/ accessed on 29 December 2022). The target genes (log2(TPM+1)) were analyzed using the DepMap web portal based on expression public 22Q4 and filtering for the COAD lineage subtype.

### 2.3. Investigation of CRC Immune-Related Genes

We used the cBioPortal for Cancer Genomics (https://www.cbioportal.org/ accessed on 29 December 2022) database to select 594 COAD patients/samples (TCGA, Pan Cancer Atlas). We selected genomic profiles based on mutations, structural variants, putative CNAs from GISTIC 2.0, mRNA expression with z-scores relative to all samples (log RNA-seq V2 RSEM), and protein expression with z-scores by reverse-phase protein array (RPPA). We then entered *HDAC2*, *B2M*, and *CIITA* and submitted the query. We used cBioPortal OncoPrint to analyze the genes of interest; we further only selected patients harboring two or three simultaneous deregulations of the genes of interest and applied a co-occurrence analysis in the queried samples, based on Fisher′s exact test.

### 2.4. Gene Ontology Enrichment

Gene ontology (GO) analysis (http://geneontology.org/ accessed on 29 December 2022) was performed using combined and individual CNA genes from the *HDAC2*, *B2M*, and *CIITA* dataset downloaded from a virtual study of altered samples (https://www.cbioportal.org/study/summary?id=63aabe8bd9f4971d767cb0be accessed on 29 December 2022). To improve the robustness of the whole analysis framework, we also compared our analysis with a wild-type group of samples (https://www.cbioportal.org/study/summary?id=63fc6e901cec6922c4239533 accessed on 29 December 2022). We explored the cellular components involved in selected CRC samples using GO cellular component as annotation dataset from the PANTHER database with a default output sorted by hierarchy of the categories. We included the name of the annotation data category, the number of genes in the reference list that map to the annotation data category, the number of genes in the uploaded list that map to the annotation data category, the number of genes expected in the list for this category, the fold enrichment, the representation grade expressed as + or -, *p*-value, and the false discovery rate (FDR).

### 2.5. Tissue Microarray Construction

All selected specimens were from formalin-fixed and paraffin-embedded tissues, and different areas were selected from hematoxylin- and eosin-stained sections, including tumor, non-neoplastic colonic mucosa, and adenomatous dysplastic modification areas. We built a TMA using cores that were representative of all three components. Three cores with a thickness of 1.0 mm were collected from each tumor sample using a Beecher TMA instrument and were inserted in a donor block. Each donor block was stained with the antibodies described in Section 2.6.

### 2.6. Immunohistochemistry Analysis

Immunohistochemical staining was carried out as described in [25]. Primary antibodies used were as follows: HDAC2 (dilution 1:250, Santa Cruz Biotechnology, Dallas, TX, USA; #sc-55541), CIITA (dilution 1:300, Abcam, Cambridge, UK; #ab117598), and B2M (dilution 1:250, Cell Signaling, Danvers, MA, USA; #9899).

### 2.7. Statistical Analysis

#### 2.7.1. Immune Infiltrate Analysis

We used the TIMER 2.0 database (http://timer.comp-genomics.org/ accessed on 29 December 2022) for tumor immune estimation, which is based on immunedeconv, an R package integrated with six different algorithms. The TIMER algorithm was used for its tissue specificity-dependent analysis. However, for *HDAC2*, we also used xCell to estimate a higher number of immune cell populations. Scatterplots were generated following the submission of input genes, showing the purity-corrected partial Spearman’s rho value and statistical significance, as well as gene expression levels against tumor purity, using microarray expression values of GBM/OV for calculation of the studied gene. The SCNA module was used to compare tumor infiltration levels among tumors with different SNCAs for the selected gene. The infiltration level for each SCNA category was compared with the normal level using a two-sided Wilcoxon rank-sum test. The mutation module was used to simultaneously analyze the effect of gene mutations on immune cell infiltration in COAD samples (n = 404) and immune cell types. Violin plots of immune infiltration distribution in mutant vs. wild-type COAD were analyzed by applying a Log2 transformation (fold change) and considering only significant data with *p* < 0.05. The outcome module was corrected for multiple covariates in a multivariable Cox proportional hazard model. Covariates included age and gene expression. Kaplan–Meier (KM) curves for the corresponding immune infiltrates and cancer types were used. The infiltration level was divided into low and high, the hazard ratio and *p*-value were used for the Cox model, while the log-rank *p*-value was used for KM analysis. Two-sided *p*-values < 0.05 were used to indicate statistical significance. Z-scores were used to indicate increased risk with *p* < 0.05 and z > 0, while decreased risk was indicated with *p* < 0.05 and z < 0. Not significant values were indicated with *p* > 0.05.

#### 2.7.2. Genomic Profile Analysis

We used the cBioPortal database (https://www.cbioportal.org/ accessed on 29 December 2022) to obtain an overview of the genomic profiles of *HDAC2*, *CIITA*, and *B2M* genes based on mutations, structural variants, putative CNAs from GISTIC, mRNA expression by applying a z-score threshold ±2 (Log RNA-seq V2 RSEM), and protein expression measured by RPPA, with a z-score ±2 relative to all samples. Mutual exclusivity was applied on the three queried genes, which resulted in an alteration in 21% of all cases, by applying a Log2 odds ratio and considering only those with a q-value (derived from Benjamini–Hochberg FDR correction procedure) cut-off < 0.05 as significant.

#### 2.7.3. Gene Ontology Enrichment Analysis

We considered only those categories with *p*-value > 0.05. The FDR was calculated using the Benjamini–Hochberg procedure. The raw *p*-values were determined by Fisher’s exact test. All results were considered valid for an overall FDR < 0.05. *p*-values were calculated by the binomial statistic.

#### 2.7.4. Tissue Microarray Analysis

The stratification of high/low expression was based on the median percentage of cells expressing the proteins. Non-parametric tests were used to compare independent groups of numerical data. Differences in the expression of *HDAC2* according to age, gender, location, and CRC category were analyzed using the Mann–Whitney U-test and Kruskal–Wallis test. The Pearson χ^2^ test was used to determine the correlation between *HDAC2* expression and the variables included in the study. A *p*-value < 0.05 was considered statistically significant. All tests used were two-tailed.

## 3. Results

### 3.1. Indirect Effect of HDAC2 on Tumor Microenvironment

Based on our preliminary findings, we investigated whether *HDAC2* might play a role in CRC immune (de)regulation and whether *CIITA* and *B2M* could somehow be involved. We explored tumor-infiltrating immune cells in COAD tumors from the TCGA cohort with different SCNAs in *CIITA*, *HDAC2*, and *B2M*. The box plots in Figure 1 show the distribution of each immune subset at each copy number status; the infiltration level for each SCNA category was compared with the control. We focused only on statistically significant data to better investigate the role of a specific immune subset. The distribution of CD8^+^ T cells and neutrophils in *CIITA* SCNAs was associated with a highly significant arm-level gain, while that of the dendritic cells showed a lower significance. An association between the distribution of B cells in *HDAC2* arm-level deletion and that of CD8^+^ T cells in both arm-level deletion and gain was found, indicating that HDAC2 has an immune-related role that is shared in different immune-like combinations. Arm-level deletion associated with *B2M* showed infiltration levels of CD8^+^ T cells, neutrophils, and dendritic cells, while a lower arm-level gain was detected in neutrophils rather than adjacent normal/diploid status, and a deep deletion was observed in dendritic cells.

We then analyzed B-cell immune infiltrates based on *HDAC2* CNAs. We found that a deep deletion status correlated with a lower level of B cells compared to diploid/normal status (*p* < 0.05; Figure 2A). We also analyzed the association between HDAC2-related CNAs and macrophage infiltration levels. The data obtained revealed higher levels of macrophages in the HDAC2 deep deletion status, but this finding was not statistically significant (Figure 2B).

We next analyzed the clinical outcome of tumor-associated immune B-cell subsets regulated by *HDAC2*. KM curve plots showed that an increased CRC risk was correlated with low *HDAC2* expression and high naïve and memory B-cell infiltrates (Figure 3A–C). These data indicate that *HDAC2* gene expression plays a critical role in CRC progression, and that its association with a high expression of B cells, in particular memory B cells, improves clinical outcome by providing a second layer of defense. B cells, and the antibodies that they produce, may favor cancer emergence and spread in CRC via an immunosuppression mechanism that might be activated by an immune complex or complement activation. In addition, B cells may inhibit T-cell cytokine secretion and consequently support tumor growth. This mechanism could also be regulated by epigenetic machinery involving *HDAC2*. Histone acetylation and deacetylation levels can shape gene expression patterns in response to environmental cues associated with the B-cell differentiation cascade, although this mechanism is not crucial and it is time-dependent. It is therefore not surprising that a high CRC risk is associated with low *HDAC2* expression and high B-cell infiltration (Figure 3), while in other conditions, HDAC2 deep deletion may be linked to lower levels of B-cell alteration (Figure 2A).

We also investigated the *HDAC2*-dependent protumoral role of B cells and its correlation with other immune clusters in CRC. We investigated whether an association between the transactivator *CIITA* and *B2M* could be involved in *HDAC2*-mediated regulation. We did not find any correlation between *HDAC2* and *B2M* (Appendix A), whereas we found a negative correlation between *HDAC2* and *CIITA* (Figure 4A). We evaluated whether the negative correlation between *HDAC2* and *CIITA* could also be translated in CRC cell lines (Appendix A), but we did not find any significant data. Higher levels of mutant *HDAC2* forms were associated with *CIITA*, and vice versa (Figure 4B,C). We also found that higher levels of *CIITA* were associated with *B2M* mutations (Figure 4D).

Next, we performed an analysis based on tissue specificity to understand whether the different immune infiltration levels of T cells could have an impact on the expression of the three genes analyzed. We investigated the correlation of CD8^+^/CD4^+^ T cells in *CIITA*, *HDAC2*, and *B2M*. The data indicated that *CIITA* and *BM2* expression levels positively correlated with CD8^+^ T cells (Figure 5A), while a positive correlation with CD4^+^ T cells was only found in *CIITA* (Figure 5B). This result is in line with the finding that *CIITA* drives the expression of MHC class II, which in turn, is crucial for antigen presentation to CD4^+^ T lymphocytes. The correlation was not significant for *HDAC2* (Appendix A).

Given that different tissues can induce distinct cancer-cell-intrinsic expression and create different immune environments depending on intrinsic and extrinsic factors, such as epigenetic regulation, we also analyzed *HDAC2* expression in immune infiltration using another algorithm involving a larger number of immune cells. We found that *HDAC2* expression negatively correlated with effector memory CD4^+^ T cells and central memory CD8^+^ T cells (Figure 6A,B).

The data obtained corroborated the hypothesis of *HDAC2* involvement in immune surveillance, since the inverse correlation with T-cell subgroups in COAD suggested that phenotypic and functional changes in T-cell subsets occur during CRC progression. Our data indicated that deregulated *HDAC2* expression in CRC correlates with CD4^+^ and CD8^+^ T-cell restraint.

### 3.2. Mutated HDAC2 Is Crucial for Immune Infiltration Levels in CRC

We next used xCell to investigate differential CD4^+^ Th1 and Th2 T-cell infiltration levels in COAD tumors considering the mutated forms of *CIITA*, *HDAC2*, and *B2M* (Figure 7). The data showed that differential CD4^+^ Th1 and Th2 T-cell infiltration levels in mutated *CIITA* harbored higher expression levels compared to those in wild-type CIITA (Figure 7A), while *HDAC2* and *B2M* mutated forms displayed a significant increase only in CD4^+^ Th2 T cells (Figure 7B,C). HLA class II-negative tumors containing mutations in HLA class II regulatory genes, such as *CIITA*, were found to be significantly more likely to develop microsatellite-unstable colon carcinomas [26]. The authors hypothesize that tumor progression was promoted in a setting of extensive CD4^+^ T-cell infiltration by microsatellite-unstable colon carcinoma cells that were lacking in HLA class II expression. This study shows that the MHC class II expression in tumor cells and the associated CD4^+^ T-cell responses behave differently in a variety of tumor entities as well as in tumors with a shared etiology.

The data obtained suggested that infiltration levels of CD4^+^ and CD8^+^ T cells were deregulated when the indicated genes were mutated. We hypothesize that, in a subgroup of CRC patients, *HDAC2* mutations reorganize immune-related genes via a mechanism involving *CIITA*, the mutation of which, in turn, drives the expression levels of MHC class II in cancer cells; this is followed by the stimulation of both CD4^+^ Th1 and Th2 helper cells, which are involved in cell-mediated and humoral immune response, respectively. In contrast, the increase in *HDAC2* and *B2M* mutants in CRC appeared to be associated with a differentiated CD4^+^ Th2 subpopulation, which could affect tumor promotion.

Differential infiltration levels of central and effector memory CD8^+^ T cells detected by xCell in the three analyzed genes (Figure 8) were higher in the mutated forms, indicating that the interconnection between mutated forms of the genes involved in MHC class I and MHC class II control are regulated by an *HDAC2*-mediated mechanism involving reprogramming of CD8^+^ T-cell differentiation, which contributes to cell fate decision.

### 3.3. Evaluation of Distinct Patient Subgroups Associated with HDAC2, CIITA, and BM2 Deregulation in CRC

Using a dataset of 594 CRC patients/samples from the TCGA Pan Cancer Atlas, we looked at the percentage of alterations present in *HDAC2*, *CIITA*, and *B2M* based on mutations, mRNA expression levels, structural variants, and amplifications (Figure 9). Out of a total of 122 altered samples, we found 7% of alterations in *HDAC2*, 11% in *B2M*, and 8% in *CIITA*. Clinical data for these samples (Appendix A) reveal that these alterations are associated with a higher T stage and mainly with MSS, as shown by the virtual study of the altered group (downloadable from https://www.cbioportal.org/study/summary?id=63aabe8bd9f4971d767cb0be accessed on 29 December 2022). To strengthen our findings, we also compared the altered samples with the wild-type group (472 samples). The clinical data are reported in Appendix A and can be downloaded from https://www.cbioportal.org/study/summary?id=63fc6e901cec6922c4239533 accessed on 29 December 2022. In the wild-type group, MSS was more prevalent, while the T stage was similar to the altered group.

We identified the shared alterations either between all three or between two genes. *HDAC2*-related truncating mutations were all of unknown significance and displayed a higher percentage of low mRNA expression, while alternations in *CIITA* included a higher percentage of missense mutations with unknown significance, only one splicing mutation as a putative driver coupled with low mRNA expression levels, and a truncating mutation as a putative driver. One case of amplification of *CIITA* was also detected. *B2M* displayed a higher number of genetic alterations, particularly truncating mutations.

Some of the listed alterations were found in more than one of the genes and they are likely interrelated in terms of clinical outcome. We found that the co-occurrence of selected genes is highly significant in CRC patients, suggesting that alterations in these genes are either directly or indirectly interconnected (Table 1).

As expected, a significant co-occurrence was detected only between *HDAC2* and *B2M* and between *B2M* and *CIITA* (Table 1, bold), which is in line with our previous data showing that *HDAC2* and *CIITA* do not directly interact (Figure 4A).

We then investigated whether the CNA burden could affect different immune signatures in CRC patients. CNAs related to wild-type samples are shown in Appendix A, those related to the 122 altered samples in Appendix A, and those involving only *B2M*, *CIITA*, and *HDAC2* in Appendix A. In Appendix A, a Venn diagram shows the CNAs in each group as well as those that were commonly regulated.

To better investigate the role of cellular components involved in the different CNAs in each altered gene, we performed a GO analysis to detect the most significant cellular components involved. We first analyzed the wild-type CRC samples and then compared them to the 122 altered CRC samples simultaneously harboring CNAs in the three genes. In the wild-type group, we found that the immunoglobulin complex was mainly represented (Appendix A). In the altered group, the most significant component was the GO term intracellular membrane-bounded organelle, indicating that, in these patients, these alterations generally have an impact on the cell structures and organizations involving the re-establishment of a novel CRC phenotype (Table 2).

To determine whether specific cell components could be associated with each gene of interest, we performed a GO analysis for each individual gene. We found that for *CIITA*, the most representative components were the MHC protein complexes (Table 3).

For *B2M*, we found that the most representative cell component in this cluster analysis was again, the MHC protein complexes (Table 4).

For *HDAC2*, we found a positive involvement of MHCs with a higher fold enrichment than for *CIITA* and *B2M*, suggesting that its (de)regulation in CRC is tightly involved in immune regulation and immune-related genes (Table 5).

### 3.4. Expression Levels of HDAC2 in Tissue Microarray of CRC Patients

To corroborate our in silico findings, we further analyzed *HDAC2* expression on a TMA of a cohort of CRC patients (n = 44). The main clinicopathological features of these patients are summarized in Appendix A. *HDAC2* expression was evaluated on paraffin-embedded tissue sections by immunohistochemical (IHC) analysis of CRC and healthy tissues. The data showed a very low expression of *HDAC2* in the glandular epithelium of healthy samples and a gradual increase in transitional mucosal tissue in both the proximal and distal colon. An aberrant expression of *HDAC2* was clearly visible in the tumor tissues, especially in the proximal colon (Figure 10A). Our results revealed that *HDAC2* expression was higher in the proximal (right) rather than in the distal (left) CRC, while the elevated *HDAC2* expression correlated with vascular invasion (Figure 10B,C). These findings indicate that *HDAC2* has a particular tropism for the proximal colon. Considering that right-sided CRC patients do not respond well to conventional chemotherapies but show more promising results with immunotherapies due to their high antigenic load, we hypothesize that *HDAC2* could have a role in immune regulation.

We also speculate that *HDAC2* expression may be linked to T-cell infiltration and may be characterized by high immunogenicity. For this reason, CRC tumors with these features respond well to immunotherapy. These results corroborated and strengthened our in silico findings, indicating that *HDAC2* plays a crucial role in CRC aggressiveness and immune responsiveness.

### 3.5. HDAC2 Is a Potential Player in CRC Immune Regulation

We then explored the expression levels of CIITA in the TMA of CRC patients. By an IHC analysis we found that the nuclear expression of CIITA was predominantly localized at the stromal and transitional mucosa levels. Interestingly, *CIITA* was also expressed in lymphocytes, which were intensely stained in the peritumoral area (Figure 11A). A multivariate statistical analysis indicated that *CIITA* is associated with MSS and the presence of metastatic lymph nodes (Figure 11B). Given that our in silico data showed that *CIITA* is associated with MSS, we wondered whether the expression of *B2M* could also be linked to *HDAC2*. By a multivariate comparative analysis we found that both *HDAC2* and *CIITA* positively correlated with *B2M* expression (Figure 11C), indicating a potential role for these genes in immune surveillance and their transcriptional regulation.

These results confirmed our findings from the in silico analyses based on both *HDAC2* and *CIITA* co-occurrence with *B2M* in CRC samples (Table 1). Although no direct correlation between *CIITA* and *HDAC2* was found, we speculate that an MSS right-sided colon phenotype, characterized by an immunosenescent microenvironment, might create favorable conditions for *HDAC2* functions associated with *CIITA* and *B2M*. These results further confirmed our in silico findings, showing that the simultaneous expression of *HDAC2* and B cells has an impact on the clinical outcome.

## 4. Discussion

CRC is characterized by complex molecular lesions involving several oncogenes and tumor-suppressor genes [27]. Genetic mutations alone are not sufficient to explain the mechanism of CRC progression, and a subset of epigenetic alterations are thought to play a functional role. An evaluation of the colon cancer epigenome [28] revealed that all CRCs are characterized by aberrant DNA methylation, resulting in the deregulated expression of oncogenes and tumor suppressors, altered microRNA expression, and histone modifications [29]. Immune regulation also plays a crucial role, mostly in therapy-resistant CRC [30,31]. Tumors can be classified as “hot” if they are characterized by abundant T-cell infiltration and high accumulation of several mutations stimulating the production of aberrant neoantigens. This makes the tumor more vulnerable to recognition by the immune system, which is then able to elicit a strong immune response. In contrast, so-called cold tumors are more resistant to immunotherapy and they produce a weak immune response, with a low mutational status and low MHC class I expression. Immune-excluded tumors and immune-desert tumors are cold tumors. In the former, CD8^+^ lymphocytes do not efficiently infiltrate the tumor but they localize only marginally, while in the latter, CD8^+^ lymphocytes are absent [32,33]. Since immunotherapeutic intervention depends heavily on this type of classification, it is crucial to determine which immunological category the tumor belongs to.

Here, we attempted to identify a genetic/epigenetic/immunological CRC signature by investigating CRC samples from the TCGA database that harbor at least two or three deregulations of *CIITA*, *HDAC2*, and *B2M* genes. An infiltration level analysis in CRC patients with alterations in *HDAC2* revealed a distinct immune signature associated with lower B-cell and higher macrophage counts, indicating that the presence of *HDAC2* could be marginally involved in B-cell development in CRC. This implies that *HDAC2* has a role in the modulation of non-histone proteins associated with factors regulating immune cell transcription, which in turn, also regulate key genes such as *B2M* and *CIITA*, which are involved in MHC class I and II, respectively.

In our study, about 5% of profiled CRC patients showed low expression levels of *HDAC2*, thus affecting immunological mediators. However, some of the analyzed patients also harbored alterations in *B2M* and *CIITA* genes, resulting in a further deregulation of the immune signature. We found that lower *HDAC2* expression alone played a key role in B-cell development and macrophage polarization, and this was also reflected in a worse prognosis in CRC patients exhibiting these characteristics. We also observed a different correlation between *HDAC2*, *CIITA*, and *B2M* in terms of gene mutations, as higher levels of *HDAC2* mutant forms were associated with *CIITA* expression, and vice versa. We also found that higher levels of *CIITA* correlated with *B2M* mutations. In terms of T-cell immune infiltration, we observed that while *CIITA* and *B2M* expression was associated with CD8^+^ T cells, a positive correlation was only found between CD4^+^ T cells and *CIITA*, while *HDAC2* correlated negatively with a specific subgroup of effector memory CD4^+^ and central memory CD8^+^ T cells. These findings indicate that *HDAC2* can act as a bridge between the regulatory networks of CD4^+^ and CD8^+^ cell differentiation. The *HDAC2* mutated form, associated with *B2M*, showed a significant increase only in the CD4^+^ Th2 subgroup. Furthermore, mutated *CIITA* and *B2M* appeared to be regulated by *HDAC2*-mediated mechanisms, as the central and effector memory CD8^+^ T cells were increased in CRC patients. We also found a very significant correlation between the expression of *HDAC2* and *B2M*, while the association between *HDAC* 2 and *CIITA* was less significant, indicating that there is an indirect correlation between these two genes. Indeed, GO analyses revealed that the main regulators of MHC class I and II were highly expressed in CRC patients presenting *HDAC2* deregulations, suggesting that this regulator plays a fundamental role in the programming and differentiation of these immune complexes through mechanisms that might be explained by *HDAC2*-mediated posttranscriptional deacetylase activities. In addition, the results of our TMA analyses of a cohort of CRC patients strengthened our in silico data, as we found that high levels of *HDAC2* expression were related to tumor tissues mainly localized in the proximal colon and with greater vascular invasion. In addition, the expression of *CIITA* in the 44 TMA-analyzed patients was mainly localized in the stromal compartment, in the transitional mucosa, and at the lymphocyte level. Both *CIITA* and *HDAC2* expression levels positively correlated with those of *B2M,* confirming that the tight relationship between these three genes has an impact on immune system regulation. However, although *HDAC2* and *CIITA* may be involved in epigenetic reprogramming, further investigations are needed to map these landscapes and their specific roles.

This study has a number of limitations, since no orthogonal experiments were conducted to corroborate some of the in silico results. First of all, we did not investigate different cell infiltrates involved in the three genes of interest, but we focused on *HDAC2* in the context of *CIITA* and *B2M* based on a small number of CRC TMA samples. Furthermore, we did not perform a comparative analysis based on the mutated forms of the genes, since these data are not available from the clinicopathological information. Although the higher *HDAC2* expression associated with CRC found in the TMA analysis is not always reflected in our in silico results, we hypothesize that *HDAC2* acts as a double-edged sword depending on its expression and the immunological context. The TMA data presented here are not sufficient to definitively demonstrate this hypothesis. However, based on our findings and the increasing body of evidence in the literature, we believe that our datasets could provide useful supporting information. The results presented here may help to define a more precise immunological signature of patients affected by CRC, highlighting the role of the tumor microenvironment and extending our knowledge of different immunological signatures involving alterations in complexes driven by *CIITA* and *B2M*.

## 5. Conclusions

Our multi-approach comparative analysis could be useful in the exploration of *HDAC2* deregulation as a cancer driver. Although it is challenging to identify a direct cancer-driving criterion, many hurdles have already been overcome thanks to the combined use of ever-advancing technologies, which are able to translate research from the bench to the bedside, and multi-omics big data that are available on public platforms. Previous reports identified the truncating mutation of *HDAC2* as conferring resistance to CRC therapy. To date, however, *HDAC2* deregulations in CRC progression have not been investigated as drivers of cancer. Here, we hypothesize that alterations in *HDAC2* expression might be indirectly regulated and might themselves (de)regulate the immune surveillance and response during CRC progression. Further investigations into the role of HDAC2 deregulation in CRC could provide more conclusive evidence and pave the way toward the use of *HDAC2*-targeted therapies in the treatment of the disease.

## Figures and Tables

**Figure 1 cancers-15-01960-f001:**
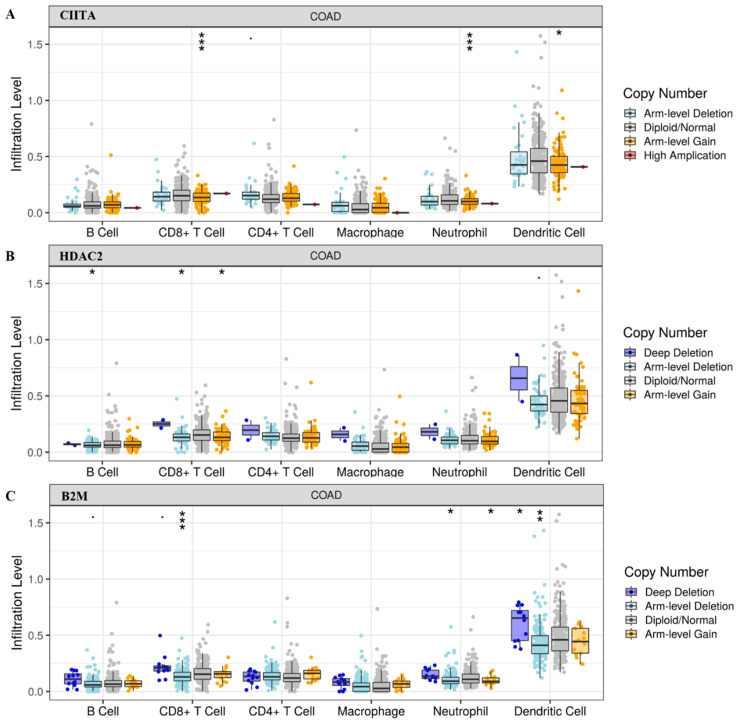
Tumor infiltration levels in (**A**) *CIITA*, (**B**) *HDAC2*, and (**C**) *B2M* in COAD tumors with SCNAs in the indicated genes. Significant *p*-values: *** ≤ 0; ** ≤ 0.001; * ≤ 0.01.

**Figure 2 cancers-15-01960-f002:**
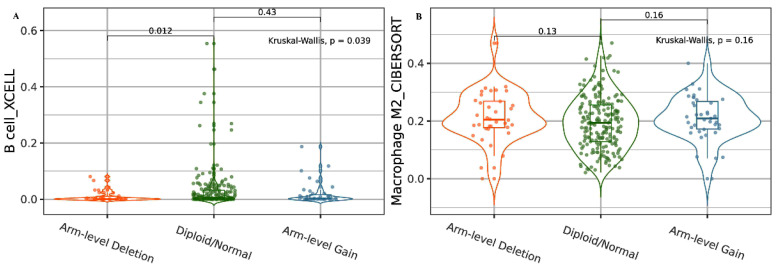
HDAC2 CNAs in B cells and macrophage immune infiltrates. (**A**) Violin plot showing the association between HDAC2 deep deletion and lower levels of B-cell alteration. (**B**) Violin plot showing the association between HDAC2 deep deletion and higher levels of M2 macrophage alteration. Data are shown in Log2 scale (fold change): higher level in mutants = *p* < 0.05, Log2FC > 0; lower level in mutants = *p* < 0.05, Log2FC < 0; not significant = *p* > 0.05.

**Figure 3 cancers-15-01960-f003:**
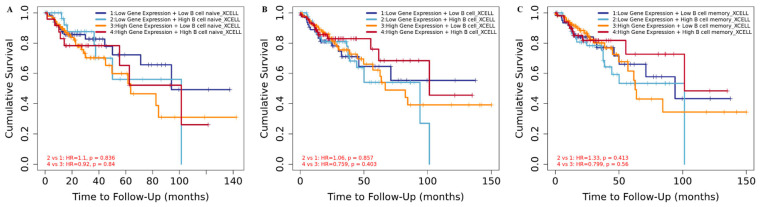
Clinical outcome of *HDAC2*-related COAD based on B-cell infiltrates. (**A**) Kaplan–Meier curve for naïve B cells in *HDAC2*-related COAD patients; score (log-rank) test *p* = 3.15 × 10^−5^ (**B**) Kaplan–Meier curve for B cells in *HDAC2*-related COAD patients; score (log-rank) test *p* = 8.86 × 10^−3^; (**C**) Kaplan–Meier curve for memory B cells in *HDAC2*-related COAD patients; score (log-rank) test *p* = 9.65 × 10^−3^.

**Figure 4 cancers-15-01960-f004:**
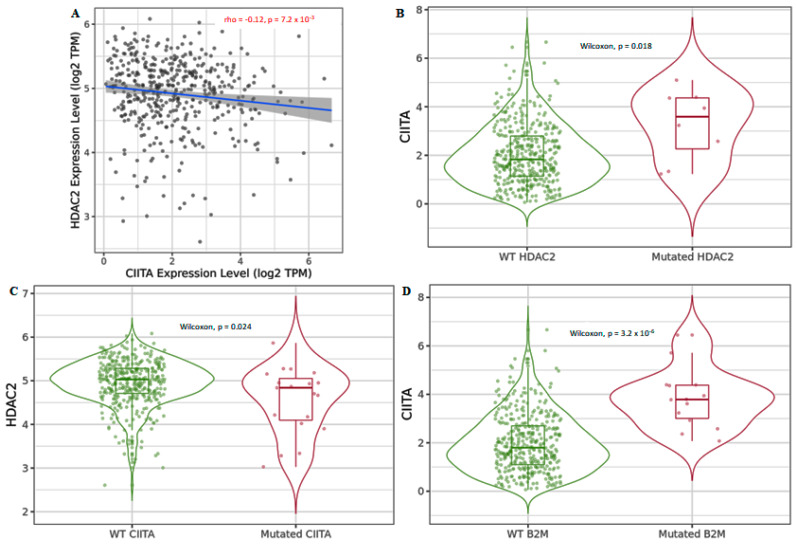
*HDAC2* and *CIITA* are negatively regulated in COAD tumors. (**A**) Correlation between *HDAC2* and *CIITA*; data shown have been purity-adjusted. Spearman’s rho value and statistical significance: positive correlation = *p* < 0.05, ρ > 0; negative correlation = *p* < 0.05, ρ < 0; not significant = *p* > 0.05; (**B**) Violin plot showing the differential expression of *CIITA* and wild-type *HDAC2* versus the mutated *HDAC2* form; (**C**) violin plot showing the differential expression of *HDAC2* and wild-type *CIITA* versus the mutated *CIITA* form; (**D**) violin plot showing the differential expression of *CIITA* and wild-type *B2M* versus the mutated *B2M* form. Data shown have been purity-adjusted. Data are shown in Log2 scale (fold change): higher level in mutants = *p* < 0.05, Log2FC > 0; lower level in mutants = *p* < 0.05, Log2FC < 0; not significant = *p* > 0.05.

**Figure 5 cancers-15-01960-f005:**
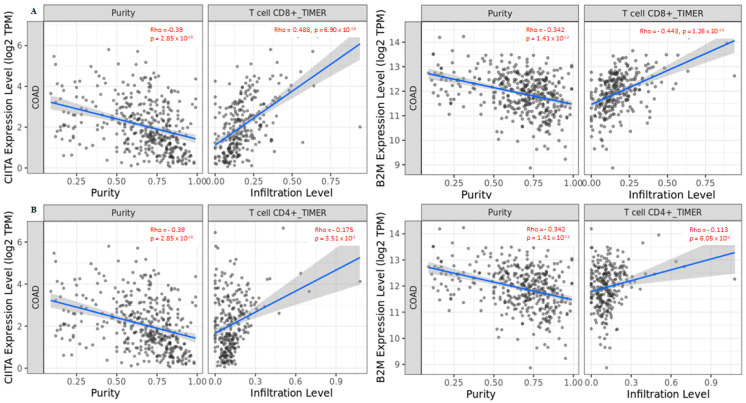
Correlation between *CIITA* and *B2M* in (**A**) CD8^+^ T cells and (**B**) CD4^+^ T cells in COAD tumors. Data shown have been purity-adjusted. Spearman’s rho value and statistical significance: positive correlation = *p* < 0.05, ρ > 0; negative correlation = *p* < 0.05, ρ < 0; not significant = *p* > 0.05.

**Figure 6 cancers-15-01960-f006:**
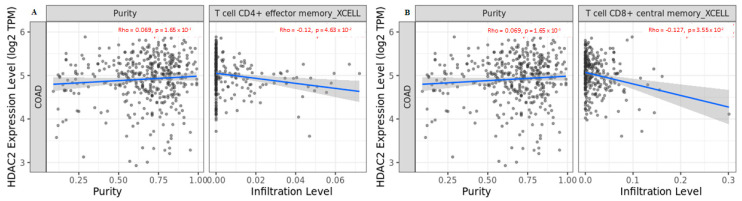
Correlation of *HDAC2* expression with (**A**) effector memory CD4^+^ T cells and (**B**) central memory CD8^+^ T cells in COAD tumors. Data shown have been purity-adjusted. Spearman’s rho value and statistical significance: positive correlation = *p* < 0.05, ρ > 0; negative correlation = *p* < 0.05, ρ < 0; not significant = *p* > 0.05.

**Figure 7 cancers-15-01960-f007:**
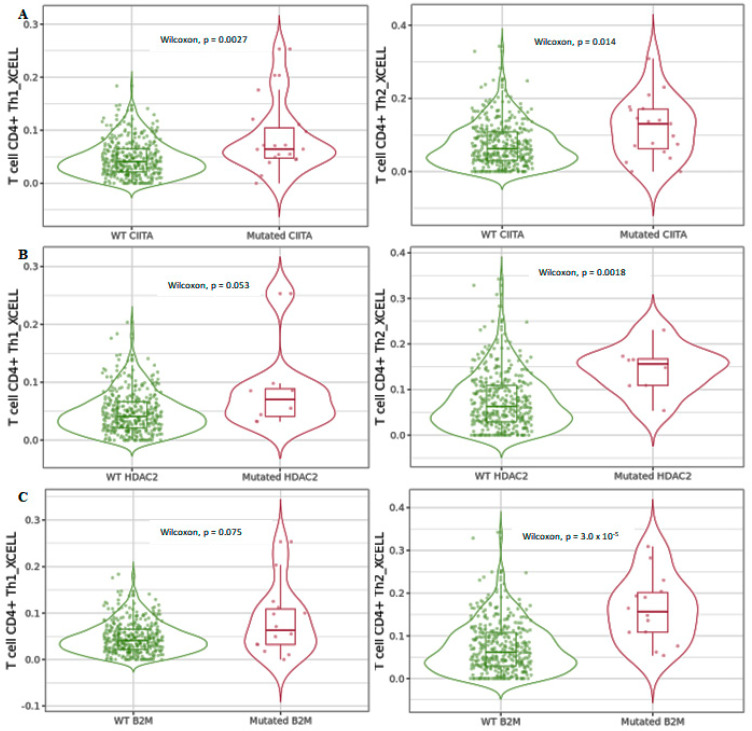
Differential CD4^+^ Th1 and Th2 T-cell infiltration levels using xCell in mutated (**A**) *CIITA*, (**B**) *HDAC2*, and (**C**) *B2M* compared to wild-type forms in COAD tumors. Data shown have been purity-adjusted. Data are shown in Log2 scale (fold change): higher level in mutants = *p* < 0.05, Log2FC > 0; lower level in mutants = *p* < 0.05, Log2FC < 0; not significant = *p* > 0.05.

**Figure 8 cancers-15-01960-f008:**
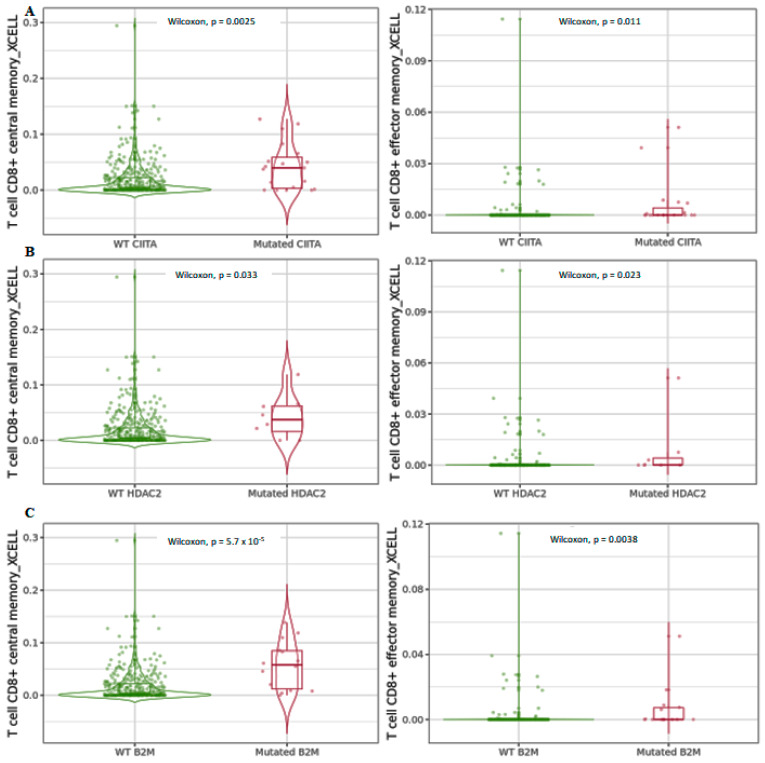
Differential central and effector memory CD8^+^ T-cell infiltration levels using xCell in mutated (**A**) *CIITA*, (**B**) *HDAC2*, and (**C**) *B2M* compared to wild-type forms in COAD tumors. Data are shown in Log2 scale (fold change): higher level in mutants = *p* < 0.05, Log2FC > 0; lower level in mutants = *p* < 0.05, Log2FC < 0; not significant = *p* > 0.05.

**Figure 9 cancers-15-01960-f009:**
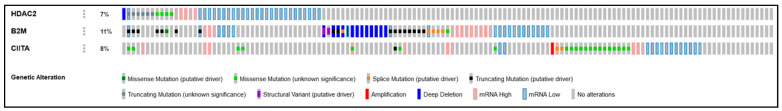
OncoPrint dataset showing *HDAC2*, *B2M*, and *CIITA* profiles in 594 CRC patients/samples harboring alterations related to the percentage of mutations, deep deletions, mRNA expression levels, structural variants, and amplifications.

**Figure 10 cancers-15-01960-f010:**
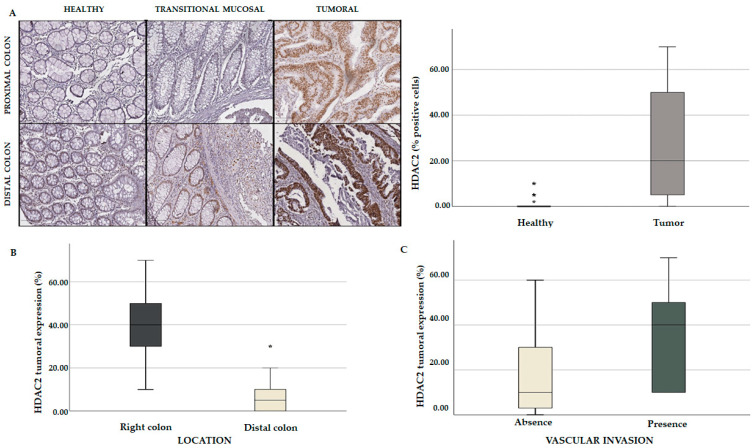
Comparative analysis of TMA containing a cohort of CRC patients. (**A**) Immunohistochemical (IHC) analysis showing *HDAC2* expression in healthy, tumoral, and transitional mucosal tissues (*p* = 0.01); *HDAC2* tumoral expression based on colon localization (*p* = 0.001) (**B**) and vascular invasion (*p* = 0.041) (**C**). Each value that deviates from the central trend of the distribution is represented in the graph with the symbol *. Values that come out of the whiskers, i.e., that are further from the box by more than 1.5 * interquartile range (IQR) up or down, are considered potential outliers and represented with symbol.

**Figure 11 cancers-15-01960-f011:**
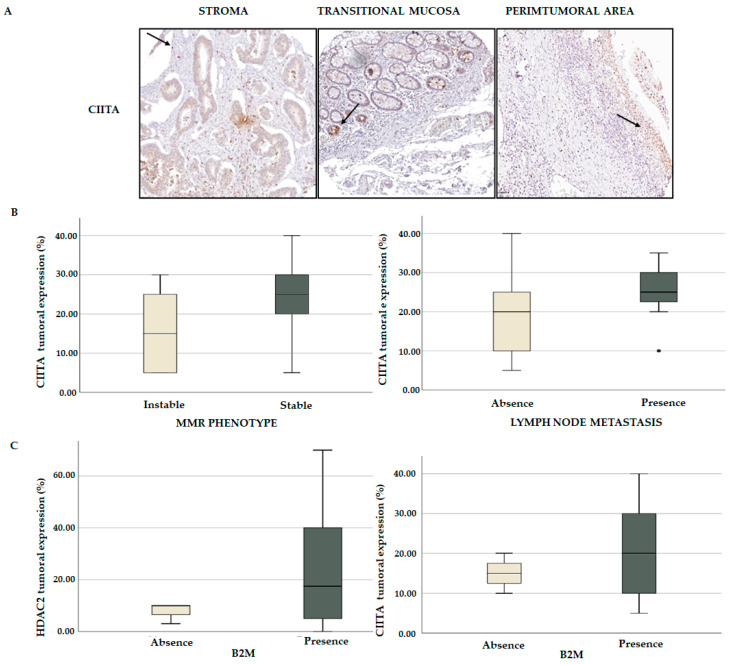
Immune regulatory players associated with HDAC2 expression. (**A**) IHC showing CIITA localization in stroma, transitional mucosa, and peritumoral area (black arrows). (**B**) CIITA tumoral expression in mismatch repair (MRR) phenotype (*p* = 0.024) and metastatic lymph nodes (*p* = 0.036). (**C**) Correlation between HDAC2 and B2M expression (*p* = 0.368); correlation between CIITA and B2M (*p* = 0.321).

**Table 1 cancers-15-01960-t001:** Co-occurrence analysis of alterations in *HDAC2*, *CIITA*, and *BM2* genes.

Co-Occurrence Genes	*p*-Value	Log Odds Ratio	Tendency
**HDAC2 and B2M**	**<0.001**	**2.277**	**Co-occurrence**
**B2M and CIITA**	**<0.001**	**2.038**	**Co-occurrence**
HDAC2 and CIITA	0.112	1.046	Co-occurrence

A total of 594 COAD patients/samples were investigated. Co-occurrence was considered significant with a *p*-value < 0.001 (**bold**).

**Table 2 cancers-15-01960-t002:** GO cellular components in 122 CRC patients with CNAs in *HDAC2*, *B2M*, and *CIITA*.

GO Cellular Component Complete	Homo Sapiens—REFLIST (20589)	Upload_1 (7435)	Expected	Over/Under	Fold Enrichment	Raw *p*-Value	FDR
intracellular membrane-bounded organelle (GO:0043231)	12,154	4634	4388.99	+	1.06	6.69 × 10^−7^	1.95 × 10^−4^
intracellular organelle (GO:0043229)	13,254	5025	4786.22	+	1.05	6.14 × 10^−7^	2.09 × 10^−4^
membrane-bounded organelle (GO:0043227)	13,230	5003	4777.55	+	1.05	2.46 × 10^−6^	6.28 × 10^−4^
organelle (GO:0043226)	14,064	5314	5078.72	+	1.05	3.71 × 10^−7^	1.52 × 10^−4^
intracellular anatomical structure (GO:0005622)	14,945	5640	5396.87	+	1.05	3.77 × 10^−8^	1.92 × 10^−5^
cytoplasm (GO:0005737)	12,097	4565	4368.41	+	1.05	6.82 × 10^−5^	1.55 × 10^−2^
plasma membrane signaling receptor complex (GO:0098802)	330	76	119.17	−	0.64	2.27 × 10^−5^	4.64 × 10^−2^
immunoglobulin complex (GO:0019814)	188	5	67.89	−	0.07	2.57 × 10^−19^	5.25 × 10^−16^
immunoglobulin complex, circulating (GO:0042571)	94	2	33.94	−	0.06	1.34 × 10^−10^	9.09 × 10^−8^
T-cell receptor complex (GO:0042101)	149	3	53.81	−	0.06	1.94 × 10^−16^	1.98 × 10^−13^

**Table 3 cancers-15-01960-t003:** GO cellular components in 122 CRC patients with CNAs in *CIITA*.

GO Cellular Component Complete	Homo Sapiens—REFLIST (20589)	Upload_1 (2227)	Expected	Over/Under	Fold Enrichment	Raw *p*-Value	FDR
MHC class I protein complex (GO:0042612)	9	8	0.97	+	8.22	8.81 × 10^−5^	9.47 × 10^−3^
MHC protein complex (GO:0042611)	26	22	2.81	+	7.82	1.18 × 10^−10^	2.41 × 10^−7^
MHC class II protein complex (GO:0042613)	18	15	1.95	+	7.70	1.25 × 10^−7^	5.10 × 10^−5^
MHC class I peptide loading complex (GO:0042824)	9	7	0.97	+	7.19	4.32 × 10^−4^	3.15 × 10^−2^
integral component of luminal side of endoplasmic reticulum membrane (GO:0071556)	29	19	3.14	+	6.06	4.18 × 10^−8^	2.84 × 10^−5^
luminal side of endoplasmic reticulum membrane (GO:0098553)	29	19	3.14	+	6.06	4.18 × 10^−8^	2.13 × 10^−5^
Golgi cis cisterna (GO:0000137)	32	17	3.46	+	4.91	1.92 × 10^−6^	3.56 × 10^−4^
luminal side of membrane (GO:0098576)	36	19	3.89	+	4.88	5.19 × 10^−7^	1.33 × 10^−4^
nucleosome (GO:0000786)	99	33	10.71	+	3.08	3.13 × 10^−7^	1.06 × 10^−4^
DNA packaging complex (GO:0044815)	146	40	15.79	+	2.53	1.43 × 10^−6^	2.93 × 10^−4^
integral component of endoplasmic reticulum membrane (GO:0030176)	169	46	18.28	+	2.52	3.29 × 10^−7^	9.61 × 10^−5^
intrinsic component of endoplasmic reticulum membrane (GO:0031227)	177	46	19.15	+	2.40	1.05 × 10^−6^	2.39 × 10^−4^
protein–DNA complex (GO:0032993)	192	47	20.77	+	2.26	2.86 × 10^−6^	4.88 × 10^−4^
endocytic vesicle membrane (GO:0030666)	195	42	21.09	+	1.99	1.49 × 10^−4^	1.33 × 10^−2^
transport vesicle (GO:0030133)	429	76	46.40	+	1.64	1.38 × 10^−4^	1.34 × 10^−2^
side of membrane (GO:0098552)	711	113	76.90	+	1.47	2.01 × 10^−4^	1.71 × 10^−2^
cell surface (GO:0009986)	995	148	107.62	+	1.38	3.35 × 10^−4^	2.73 × 10^−2^
organelle subcompartment (GO:0031984)	1501	209	162.35	+	1.29	5.08 × 10^−4^	3.58 × 10^−2^
organelle membrane (GO:0031090)	3688	466	398.91	+	1.17	5.79 × 10^−4^	3.94 × 10^−2^
intracellular membrane-bounded organelle (GO:0043231)	12,154	1423	1314.63	+	1.08	8.36 × 10^−6^	1.14 × 10^−3^
membrane-bounded organelle (GO:0043227)	13,230	1537	1431.02	+	1.07	7.31 × 10^−6^	1.07 × 10^−3^
intracellular organelle (GO:0043229)	13,254	1534	1433.61	+	1.07	2.11 × 10^−5^	2.54 × 10^−3^
organelle (GO:0043226)	14,064	1622	1521.23	+	1.07	1.05 × 10^−5^	1.34 × 10^−3^
intracellular anatomical structure (GO:0005622)	14,945	1716	1616.52	+	1.06	5.31 × 10^−6^	8.35 × 10^−4^
cellular anatomical entity (GO:0110165)	18,802	2085	2033.71	+	1.03	1.37 × 10^−4^	1.40 × 10^−2^
cellular component (GO:0005575)	18,949	2096	2049.61	+	1.02	3.87 × 10^−4^	3.04 × 10^−2^
Unclassified (UNCLASSIFIED)	1640	131	177.39	−	0.74	3.87 × 10^−4^	2.93 × 10^−2^
T cell receptor complex (GO:0042101)	149	2	16.12	−	0.12	4.65 × 10^−5^	5.28 × 10^−3^
immunoglobulin complex (GO:0019814)	188	0	20.33	−	<0.01	7.16 × 10^−9^	7.32 × 10^−6^
immunoglobulin complex, circulating (GO:0042571)	94	0	10.17	−	<0.01	1.45 × 10^−4^	1.34 × 10^−2^

**Table 4 cancers-15-01960-t004:** GO cellular components in 122 CRC patients with CNAs in *B2M*.

GO Cellular Component Complete	Homo Sapiens—REFLIST (20589)	Upload_1 (3327)	Upload_1 (Expected)	Upload_1 (Over/Under)	Upload_1 (Fold Enrichment)	Upload_1 (Raw *p*-Value)	Upload_1 (FDR)
MHC protein complex (GO:0042611)	26	22	4.20	+	5.24	9.31 × 10^−8^	4.76 × 10^−5^
MHC class II protein complex (GO:0042613)	18	15	2.91	+	5.16	1.18 × 10^−5^	2.01 × 10^−3^
integral component of luminal side of endoplasmic reticulum membrane (GO:0071556)	29	19	4.69	+	4.05	1.00 × 10^−5^	2.05 × 10^−3^
luminal side of endoplasmic reticulum membrane (GO:0098553)	29	19	4.69	+	4.05	1.00 × 10^−5^	1.86 × 10^−3^
luminal side of membrane (GO:0098576)	36	20	5.82	+	3.44	3.52 × 10^−5^	5.53 × 10^−3^
ER to Golgi transport vesicle membrane (GO:0012507)	62	25	10.02	+	2.50	2.71 × 10^−4^	3.25 × 10^−2^
anchored component of membrane (GO:0031225)	171	51	27.63	+	1.85	2.72 × 10^−4^	3.08 × 10^−2^
extracellular region (GO:0005576)	4396	806	710.35	+	1.13	2.22 × 10^−4^	2.84 × 10^−2^
endomembrane system (GO:0012505)	4749	867	767.40	+	1.13	1.79 × 10^−4^	2.43 × 10^−2^
cytoplasm (GO:0005737)	12,097	2100	1954.77	+	1.07	1.96 × 10^−6^	5.72 × 10^−4^
organelle (GO:0043226)	14,064	2434	2272.62	+	1.07	1.52 × 10^−8^	1.55 × 10^−5^
intracellular membrane-bounded organelle (GO:0043231)	12,154	2102	1963.98	+	1.07	5.81 × 10^−6^	1.32 × 10^−3^
intracellular organelle (GO:0043229)	13,254	2292	2141.73	+	1.07	3.48 × 10^−7^	1.19 × 10^−4^
membrane-bounded organelle (GO:0043227)	13,230	2272	2137.85	+	1.06	5.62 × 10^−6^	1.43 × 10^−3^
intracellular anatomical structure (GO:0005622)	14,945	2555	2414.98	+	1.06	2.94 × 10^−7^	1.20 × 10^−4^
cellular component (GO:0005575)	18,949	3121	3061.99	+	1.02	3.07 × 10^−4^	3.30 × 10^−2^
unclassified (UNCLASSIFIED)	1640	206	265.01	−	0.78	3.07 × 10^−4^	3.13 × 10^−2^
immunoglobulin complex (GO:0019814)	188	4	30.38	−	0.13	2.35 × 10^−8^	1.60 × 10^−5^
immunoglobulin complex, circulating (GO:0042571)	94	2	15.19	−	0.13	1.53 × 10^−4^	2.24 × 10^−2^
T-cell receptor complex (GO:0042101)	149	1	24.08	−	0.04	7.76 × 10^−9^	1.59 × 10^−5^

**Table 5 cancers-15-01960-t005:** GO cellular components in 122 CRC patients with CNAs in *HDAC2*.

GO Cellular Component Complete	Homo Sapiens—REFLIST (20589)	Upload_1 (1614)	Expected	Over/Under	Fold Enrichment	Raw *p*-Value	FDR
MHC protein complex (GO:0042611)	26	21	2.04	+	10.30	2.17 × 10^−12^	4.44 × 10^−9^
MHC class II protein complex (GO:0042613)	18	14	1.41	+	9.92	1.47 × 10^−8^	6.02 × 10^−6^
MHC class I protein complex (GO:0042612)	9	7	0.71	+	9.92	6.74 × 10^−5^	7.25 × 10^−3^
integral component of luminal side of endoplasmic reticulum membrane (GO:0071556)	29	17	2.27	+	7.48	9.33 × 10^−9^	6.36 × 10^−6^
luminal side of endoplasmic reticulum membrane (GO:0098553)	29	17	2.27	+	7.48	9.33 × 10^−9^	4.77 × 10^−6^
luminal side of membrane (GO:0098576)	36	17	2.82	+	6.02	1.06 × 10^−7^	3.60 × 10^−5^
nucleosome (GO:0000786)	99	30	7.76	+	3.87	9.24 × 10^−9^	9.44 × 10^−6^
ER to Golgi transport vesicle membrane (GO:0012507)	62	17	4.86	+	3.50	4.28 × 10^−5^	5.14 × 10^−3^
DNA packaging complex (GO:0044815)	146	34	11.45	+	2.97	2.37 × 10^−7^	6.91 × 10^−5^
integral component of endoplasmic reticulum membrane (GO:0030176)	169	36	13.25	+	2.72	8.57 × 10^−7^	1.95 × 10^−4^
intrinsic component of endoplasmic reticulum membrane (GO:0031227)	177	36	13.88	+	2.59	1.83 × 10^−6^	3.75 × 10^−4^
protein–DNA complex (GO:0032993)	192	39	15.05	+	2.59	6.97 × 10^−7^	1.78 × 10^−4^
endocytic vesicle membrane (GO:0030666)	195	33	15.29	+	2.16	1.52 × 10^−4^	1.47 × 10^−2^
early endosome membrane (GO:0031901)	185	31	14.50	+	2.14	2.90 × 10^−4^	2.58 × 10^−2^
endosome membrane (GO:0010008)	544	68	42.64	+	1.59	4.79 × 10^−4^	3.91 × 10^−2^
lytic vacuole (GO:0000323)	749	88	58.72	+	1.50	5.09 × 10^−4^	4.00 × 10^−2^
lysosome (GO:0005764)	749	88	58.72	+	1.50	5.09 × 10^−4^	3.85 × 10^−2^
vacuole (GO:0005773)	845	97	66.24	+	1.46	5.10 × 10^−4^	3.72 × 10^−2^
chromatin (GO:0000785)	1278	146	100.18	+	1.46	1.83 × 10^−5^	2.88 × 10^−3^
chromosome (GO:0005694)	1852	191	145.18	+	1.32	2.40 × 10^−4^	2.23 × 10^−2^
nucleus (GO:0005634)	7682	680	602.20	+	1.13	1.35 × 10^−4^	1.38 × 10^−2^
intracellular membrane-bounded organelle (GO:0043231)	12,154	1039	952.77	+	1.09	2.51 × 10^−5^	3.66 × 10^−3^
organelle (GO:0043226)	14,064	1192	1102.50	+	1.08	2.78 × 10^−6^	5.17 × 10^−4^
membrane-bounded organelle (GO:0043227)	13,230	1119	1037.12	+	1.08	3.51 × 10^−5^	4.48 × 10^−3^
intracellular organelle (GO:0043229)	13,254	1119	1039.00	+	1.08	5.50 × 10^−5^	6.25 × 10^−3^
intracellular anatomical structure (GO:0005622)	14,945	1248	1171.56	+	1.07	3.14 × 10^−5^	4.27 × 10^−3^
T cell receptor complex (GO:0042101)	149	1	11.68	−	0.09	3.59 × 10^−4^	3.06 × 10^−2^
immunoglobulin complex (GO:0019814)	188	1	14.74	−	0.07	1.57 × 10^−5^	2.68 × 10^−3^

## Data Availability

The data presented in this study are available in this article (and in Appendix A). Appendix A are available online at https://www.cbioportal.org/study/summary?id=63aabe8bd9f4971d767cb0be accessed on 29 December 2022 and https://www.cbioportal.org/study/summary?id=63fc6e901cec6922c4239533 accessed on 29 December 2022.

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
