# Peer review of "Targeting HDAC2-Mediated Immune Regulation to Overcome Therapeutic Resistance in Mutant Colorectal Cancer"

_cancers, 2023, doi:10.3390/cancers15071960_

Round 1

Reviewer 1 Report

In this paper, Conte et al correlated genetic, epigenetic and micro-environment factors in colorectal cancer in order to inform novel alteration signatures to inform the clinical outcomes. I have the following critiques about the paper:

1. The selection of samples altered in the three genes under question can introduce a selection bias and a fourth comparison with total wild type group will provide more robustness to the whole analysis framework

2. The differences in B cell infiltration and the CNAs in HDAC2 are statistically marginal at best. The statement "...related immune signature subsisted and were associated with lower B cell and higher macrophage counts indicating that the presence of HDAC2 is critical for B cell development in CRC and is involved in the phenotypic conversion of M1 to M2 macrophages." (lines 441 to 444) is overstating the importance of this finding. I would suggest changing the language in the discussion.

3. The survival analyses results from  with Cox proportional hazard models are not discussed extensively in the paper. It seems those results were not significant (from the figures 3A-C). I suggest the authors expand on that. Providing log-rank p values are not sufficient given that more sophisticated models were used but not reported adequately. 

4. The correlations of all the expressions and infiltration levels, even after correcting for p-values, in Figure 5B for CIITA is driven by a few leverage points and not robust enough.

5. The authors should comment on how they dealt with mutation status in low purity samples. It would seem the mutant status in these genes will be hard to evaluate in lower purity (< 20% is canonical) samples. 

6. I appreciate the TMA analyses but a clearer explanation of the results will help make the case for the authors. 

7. Since no orthogonal experiments were conducted to corroborate some of the in-silico results, authors should address this caveat in the discussion.

8. Authors should tone down the conclusions for the same reasons.

9. Increasing the quality (resolution) of the figures will help interpret the results without struggle.

10. Please address several typos present in the text. 

Reviewer 2 Report

Targeting HDAC2-mediated immune regulation to overcome therapeutic resistance in mutant colorectal cancer

Conte M., et al.

Summary:

Conte et. al. provide data that characterize the correlation of HDAC2 expression in colorectal cancers to the expression levels of MHC class II or ß2-microglobulin genes. Extensive analyses of tumor databases reveal that the expression of CTIIA and B2M are correlated to that of HDAC2.

The study is sound, conducted well, and the data support the hypotheses put forth by the authors, and will be of interest to the community. I recommend publishing this study.

I have a few general comments:

  1. Most figures have almost illegible font sizes for axes, labels, etc. Please fix this so that the data are more easily consumed.

  2. There are multiple figures where the authors use “purity-adjusted” datasets. It would be great if they indicated how this was done, and which method was adopted for the purity adjustment.

  3. The authors present data showing that HDAC2 levels negatively correlate with levels of CTIIA in tumor samples. It would be interesting to see if/how this correlation translates to tumor cell lines. While there are differences between tumors and the cell lines derived thereof it would still be interesting, nevertheless. Perhaps the authors could mine the DepMap/CCLE databases to dwell into this.

  4. Lines 258-259: The authors indicate that they did not observe any correlation between HDAC2 and B2M levels. It would be nice to see this data, and the statistical (in)significance as a supplemental figure.

  5. How does HDAC2 expression regulate the expression of other tumor immunity-related genes? What are their correlations from tumor data-sets for HDAC2 vs major immune pathways that CTIIA and B2M are responsible for?

I would like to apologize for the delay in communicating my review of this article!

Reviewer 3 Report

The authors explored the role of three genes in CRC using their expression, mutation and CNV states. Overall analysis is interesting, but the claims in the paper lack support for the mechanisms and rely a lot on correlations which sometimes contradict based on different figures and aren’t explained well or explored. Hence some major improvements are required in manuscript as below:

1.     Please change Gene Onthology to ‘Gene Ontology’ in the manuscript everywhere.

2.     Line I88: It includes too many ‘and’; last half of the sentence doesn’t make sense

3.     Line 205: Please add ‘.’ at the end of paragraph.

4.     Expand COAD as ‘colon adenocarcinoma’ in the methods section where it first appears in the manuscript.

5.     Explain why authors looked into the arm-length deletions/gains but not the ‘focal’ deletions or gains.

6.     Line 229: It say “while an arm level gain was detected in neutrophil”. However, in Figure 1C, it seems that average or median in Neutrophil (yellow boxplot) for B2M is lower compared to the adjacent Normal/Diploid. Please explain and/or correct it.

7.     Line 238: It says “higher levels of macrophages were found in the HDAC2 deep deletion status (Figure 2B)”. However, p-values don’t support this statement i.e. P >= 0.13. Please remove this sentence and anywhere it is used to justify some claims in the rest manuscript. You can write its higher but not statistically significant in the Figure legend (Line 242).

8.     Line 236: Please mention p-values here in bracket besides Figure 2A. For example (Figure 2A, P < 0.05)

9.     Line 246-47: No experiment supports this claim. If there is one, please mention the Figure besides or remove the sentence.

10.  Line 250: Please mention why high levels of B-Cell infiltration is associated with increased CRC risk. Also, mention the disconnect where ‘HDAC2 deletion is associated with low B-Cell infiltration’, while ‘high CRC risk has low HDAC2 & high B-Cell infiltration’. Low HDAC2 should not allow for high B-Cell infiltration as per figure 2A.

11.  Line 274: Please replace ‘of our genes’ with ‘of analyzed three genes’

12.  Figure 5 and Figure 7A have a disconnect and should be explained. Why there is positive correlation in terms of expression for CIITA with infiltration, while mutations in CIITA leads to higher CD4+ T-cell infiltration? Please explain.

13.  Line 311-315: No experiment supports the claims on mechanism. If there is one, please mention the Figure besides or remove the sentence. No references were given either. Just mere correlations don’t make those claims true, experiments need to done. Otherwise please just mention that it is authors’ hypothesis or possible explanation.

14.  Line 320-21: Mentions ‘epigenetically’. Please remove it, there is no epigenetics experiment done or analysed in the manuscript. If there was any DNA methylation, ATAC-seq or histone modifications analysis done, then authors could have explained their narrative on epigenetics. Please remove it from other sections in the paper as well. It is fine to mention in the ‘discussion’ section that HDAC2 and CIITA are possibly involved in epigenetic reprogramming, however future experiments are needed to map those landscapes and their specific roles. Also, ‘transcriptome’ or transcriptionally could be used instead where needed.

15.  Line 411-412: Why only HDAC2? And same comment as above, there is need of more explanation or evidence about the statement.

Round 2

Reviewer 1 Report

No further comments